# SCALING LANGUAGE MODEL RELIABILITY VIA DETERMINANTAL POINT PROCESS PROMPT SAMPLING

## ABSTRACT

Language models achieve stronger performance when given multiple opportunities to solve a task, as in best-of-$N$ inference. However, naive approaches to scaling at test time—such as high-temperature sampling or random prompt ensembling—suffer from correlated failures, where many attempts repeat the same mistakes. We argue that improving pass@$k$ performance requires selecting prompts that are individually strong at eliciting correct answers while also nudging the model toward semantically distinct reasoning paths. To this goal, we introduce a lightweight, query-conditioned framework for prompt selection based on Determinantal Point Processes (DPPs). We build an accuracy–diversity target kernel by combining accuracy labels with hidden-activation similarities, and train a small encoder to approximate this target kernel. The encoder is optimized via a Kullback-Leibler divergence objective, which admits an unbiased gradient estimator. Given the compute budget of $k$ generations at inference, the encoder alone is used to generate the test-time DPP and sample a diverse subset of $k$ prompts that maximize coverage of complementary paths. Experiments on multiple benchmarks demonstrate that our approach outperforms competitive baselines.

## 1 INTRODUCTION

Language models are demonstrating remarkable capabilities ranging from reasoning (Imani et al., 2023), content generation (Agossah et al., 2023), to interactive conversational (Mahmood et al., 2023). However, at deployment time, a single decoding pass is often not reliable, especially for small models. In this paper, we investigate methods for enhancing reliability at inference time when the system has an additional budget of $k$ attempts. We focus on two goals that matter for deployment. The first is to increase pass@$k$ on reasoning-heavy tasks by finding at least one correct answer within $k$ trials. Second, we aim to reduce false negatives in safety-critical guardrails by covering a broader range of failure modes. This deployment-time reliability improvement complements the base model's capability: the base model provides the anchor for what is achievable, and the additional attempt budget allows us to improve reliability without modifying model parameters.

Existing approaches to use this budget at test time (Muennighoff et al., 2025; Snell et al., 2024), include BEST-OF-$K$ sampling with higher temperature (Lightman et al., 2023a), beam search (Wu et al., 2024), or search over solution trees (Zheng et al.). These methods often help large models that produce diverse outputs under a fixed prompt. However, smaller models tend to exhibit correlated failures, where many attempts repeat the same incorrect reasoning or overlook the same safety pattern, so additional generations do not result in higher pass@$k$ or lower false negatives. These limits motivate an input-side strategy that uses multiple prompts rather than multiple decodings from a single prompt. Existing works (Wang et al., 2025) explore input-prompt ensembling or perturbation, yet with typical query-agnostic or fixed heuristic rules (such as random selection). They do not reliably balance individual prompt quality and inter-prompt diversity for each input query.

Motivated by this gap, we consider a query-conditioned alternative in which, for each input, we choose a small subset of instruction prompts so that the model explores distinct reasoning or detection modes within $k$ attempts. The subset must satisfy three requirements: (i) each selected prompt should be individually effective at eliciting correct answers or correct safety decisions; (ii) taken together, the prompts should induce semantically distinct behaviors to reduce correlated failures; and (iii) the subset should be small enough to keep the added inference cost under control. The existence

of such sets is plausible as instruction prompts steer model behavior. Different instructions can shift the distribution of reasoning steps or focus on different safety cues (Li et al., 2023; Schulhoff et al., 2025; Lau et al., 2024). Moreover, modern language models act as in-context learners (Olsson et al., 2022), so instruction and few-shot exemplars can change internal representations and lead to distinct modes even when the base weights are fixed. Despite its existence, selecting such a subset from a large prompt library in a query-conditioned way is a hard problem. Random selection is fast and simple, but it performs inconsistently and ignores query-specific differences in prompt quality and overlap. We need a query-conditioned selection solution that balances quality and diversity, ideally keeps the subset small, and incurs minimal overhead relative to one generation's pass of the model.

To solve this problem, we introduce a lightweight, query-conditioned framework for prompt selection based on Determinantal Point Processes (DPPs). DPPs are probabilistic models that provide a mathematically grounded method for selecting a diverse subset of items (Liu et al., 2022; Yang et al., 2023; Hough et al., 2006; Chen et al., 2018). Our method learns to construct a kernel matrix for the DPP that balances two key factors for any given query: the individual quality of a prompt and the pairwise diversity between prompts. We train a small encoder network to approximate a target kernel matrix that combines ground-truth performance labels (e.g., accuracy for math tasks, detection success for safety tasks) with similarity measures based on the model's hidden activations. The encoder is optimized to minimize the Kullback-Leibler divergence between its predicted DPP distribution and a target distribution that embodies the ideal trade-off between quality and diversity. At inference time, for a new query, we use the trained encoder to rapidly compute a DPP kernel over all prompts and then sample a subset of $k$ prompts. This provides a principled way to select a small set of prompts that are both high-quality and diverse, which scales language model reliability. For reasoning tasks, this helps explore distinct solution pathways to find a correct answer. For guardrail tasks, this helps cover different failure modes to robustly detect harmful content.

By shifting the computational burden to an offline training phase and an efficient inference-time selection step, our method offers a scalable approach to enhance both forms of reliability without incurring the costs associated with naive prompt ensembling or the correlated failures of high-temperature sampling. Experiments on multiple benchmarks, from math and coding to safety guardrails, demonstrate that our approach consistently outperforms competitive BEST-OF-K and RANDOM-K baselines, showing that structured diversity, achieved through learned DPP sampling, is a powerful and general tool for scaling language model reliability.

**Contributions.** This paper aims to select a diverse subset of prompts from a prompt universe at test-time. By selecting a diverse set of prompts, we aim to mitigate correlated prompt failures and improve the language model's reliability. Our contributions are summarized as follows.

- **A DPP-based framework for prompt selection**. We formulate the test-time prompt selection as subset sampling from a learned, query-conditioned Determinantal Point Process (DPP). This provides a principled approach to explicitly balance prompt quality (accuracy) and diversity, directly addressing the goal of maximizing pass@$k$. Our framework is instantiated by a lightweight neural encoder that dynamically parameterizes the DPP kernel matrix for each unique query.

- **A novel supervised learning objective**. To train the encoder, we propose a new objective based on minimizing the expected KL divergence between two DPP distributions. The first is the predicted DPP, parameterized by our model's representations. The second is a target DPP, which we construct from ground-truth solution correctness labels from the training data. This objective directly trains the model to generate distributions over prompt subsets that align with empirical optimal choices.

- **An unbiased and efficient gradient estimator**. The gradient of our KL divergence loss is intractable due to an expectation over the combinatorial space of subsets. We address this by deriving an unbiased gradient estimator that uses importance sampling to make the computation feasible. This key technical contribution makes our framework tractable, enabling stable and efficient end-to-end training with standard first-order methods, such as SGD or Adam.

## 2 RELATED WORKS

**Test-time Scaling for Improved Reliability.** An established strategy for improving language model performance is to use additional computation during inference. Existing methods can be

broadly categorized based on where this computation is applied. A large body of work focuses on generating multiple, diverse output sequences from a single fixed prompt. This includes techniques like high-temperature sampling used in BEST-OF-K (Lightman et al., 2023b), beam search (Freitag & Al-Onaizan, 2017), and more structured tree-search algorithms that explore a branching space of reasoning steps (Zhou et al., 2023; Yao et al., 2024; Zhang et al., 2025). While effective for large models with high entropy outputs, these decoding-time methods often suffer from correlated failures in smaller models, which tend to produce lower-entropy, less varied outputs from a single prompt.

An alternative strategy, and the one we adopt, is to instead scale performance by using multiple, diverse input prompts while generating only a single output per prompt. This approach is more efficient for smaller models, as the cost of running a small encoder to select prompts is negligible compared to the cost of multiple full generations. The core challenge of this strategy is no longer decoding, but the combinatorial selection problem: from a large universe of candidate prompts, choose a small subset of size $k$ that maximizes the probability of success (pass@$k$). This requires high-quality and diverse selections to avoid correlated failures.

Prior work has shown that manually or automatically designed diverse prompts can improve performance (Li et al., 2023; Naik et al., 2023). Other studies have explored prompt ensembling (Lau et al., 2024) or perturbation (Wang et al., 2025). However, these methods often rely on fixed, query-agnostic strategies for selecting or constructing prompts, such as random selection or predefined rules. They do not dynamically select an optimal, query-specific subset that balances the dual objectives of individual prompt quality and inter-prompt diversity, which is necessary to maximize pass@$k$ under a fixed budget $k$.

**Diverse Subset Selection with Determinantal Point Processes.** Determinantal Point Processes (DPPs) offer a principled probabilistic framework for selecting diverse subsets. They have been widely applied in machine learning for tasks like recommendation (Liu et al., 2022) and exemplar selection for in-context learning (Yang et al., 2023), with efficient algorithms for both exact and approximate sampling (Hough et al., 2006; Chen et al., 2018; Poulson, 2020; Calandriello et al., 2020). The primary challenge in using DPPs is kernel learning. Standard methods typically involve estimating the parameters of a fixed, pre-defined kernel function from observed data (Kulesza & Taskar, 2011; Brunel et al., 2017; Anquetil et al., 2020). This approach is insufficient for our problem because the ideal kernel for prompt selection is not fixed. It is inherently query-conditioned. The quality of a prompt and its similarity to other prompts depend on the specific input question. A fixed kernel cannot capture this dynamic, context-dependent nature of quality and diversity.

Our work addresses this limitation. We move beyond fixed kernels by learning a lightweight encoder that dynamically parameterizes the DPP kernel for each unique query. This allows the selection strategy to adapt to the specific requirements of each problem instance. Furthermore, we overcome the challenge of learning this encoder by introducing a novel training objective based on the Kullback-Leibler divergence between DPP distributions, for which we derive an unbiased gradient estimator. This enables end-to-end training. Compared with prior applications of DPPs, we provide a flexible and effective solution for the query-specific prompt selection problem.

## 3 PRELIMINARIES

**Notations.** We use $\mathcal{M} = \{p_m\}_{m=1}^{M}$ to denote the universe of $M$ instruction prompts, with $[M] = \{1, \ldots, M\}$ representing the prompt index set. Questions $q$ are drawn from distribution $\mathbb{Q}$, and $x_{im} = [p_m, q_i]$ represents the concatenated prompt-question input. The hidden representation from the base model is denoted $h_{im}$, while $\phi(\cdot; \theta)$ represents our lightweight encoder with parameters $\theta$, producing feature representations $f_{im} = \phi(x_{im}; \theta) \in \mathbb{R}^d$. For each question $q_i$, we form the feature matrix $F_i = [f_{i1}, \ldots, f_{iM}] \in \mathbb{R}^{d \times M}$. Random subsets from the DPP distribution are denoted $\tilde{Y} \subseteq [M]$, and $k$ represents the inference budget (subset size). The identity matrix is denoted $I$.

### 3.1 DETERMINANTAL POINT PROCESSES

Determinantal Point Processes (DPPs) are a family of probabilistic measures that originate from the field of quantum physics, which can help us find a diverse representative subset. We begin with the $L$-ensemble definition of DPP from (Kulesza & Taskar, 2012).

**Definition 1** (Determinantal point process). *Let $L \in \mathbb{R}^{M \times M}$ be a positive semidefinite matrix, and $\tilde{Y}$ is a random subset with values in $2^{[M]}$, the set of all subsets of $[M]$. Then $\tilde{Y}$ is a determinantal point process (DPP) described by $L$ if for any set $Y \subseteq [M]$, we have*

$$\mathbb{P}_L(\tilde{Y} = Y) = \frac{\det(L_Y)}{\det(I + L)}, \tag{1}$$

*where $L_Y$ is the submatrix of $L$ indexed by the elements of $Y$.*

Hereafter, $\mathrm{DPP}(L)$ denotes a DPP defined by the ensemble matrix $L$. By convention, $\det(L_\emptyset) = 1$ and thus $\mathbb{P}_L(|\tilde{Y}| = 0) = \mathbb{P}_L(\tilde{Y} = \emptyset) = \frac{1}{\det(I+L)}$ (Kulesza, 2012, equation (4.26)). The ensemble definition is useful because it allows us to evaluate the probability that any set $Y$ is drawn. Alternatively, DPP can be equivalently characterized by the marginal kernel, thanks to the following lemma.

**Lemma 2** (Equivalent characterization). *For any $L$-ensemble matrix $L \succeq 0$, there exists a corresponding marginal kernel matrix $K = L(I + L)^{-1}$. Conversely, if $0 \preceq K \prec I$, then the associated $L$-ensemble matrix is $L = K(I - K)^{-1}$.*

The marginal kernel $K$ captures the *inclusion* probabilities for subsets of items: for any $Y \subseteq \mathcal{Y}$, we have $\mathbb{P}(Y \subseteq \tilde{Y}) = \det(K_Y)$, where $K_Y$ is the restriction of $K$ to the rows and columns indexed by $Y$. For a single item $i$, we have $\mathbb{P}(i \in \tilde{Y}) = K_{ii}$, the diagonal element of the matrix $K$.

DPPs suit our objective because they induce diversity in the sampling process via negative correlations, which can potentially mitigate correlated prompt failures. At inference time, DPPs admit an approximate maximum a posteriori (MAP) solution with cardinality $k$ via a greedy algorithm with $O(1/k!)$ approximation ratio (Civril & Magdon-Ismail, 2009).

## 3.2 Challenges in Learning DPPs

Learning representations for DPPs is particularly challenging because multiple kernel matrices could induce the same DPP (in a probabilistic manner). To see this, we first recite the notion of $D$-similarity between two square matrices (Kulesza, 2012).

**Definition 3** ($D$-similarity). *Two matrices $L_1, L_2 \in \mathbb{R}^{M \times M}$ are $D$-**similar**, if $L_2 = DL_1D$ for some diagonal matrix $D \in \mathbb{R}^{M \times M}$ with diagonal entries $D_{ii} \in \{+1, -1\}$ for $i = 1, \ldots, M$.*

Multiplying a diagonal matrix $D$ with entries in $\{+1, -1\}$ to both the left- and right-hand side of $L_1$ will keep the diagonal values the same. However, the $(i, j)$ component of the matrix could flip the sign if $D_{ii}D_{jj} = -1$. While the signs of the off-diagonal terms could be opposite, it is surprising that $L_1$ and $L_2$ could represent the same DPP in terms of distribution. This fact is highlighted in the following theorem (Kulesza, 2012, Theorem 4.1).

**Theorem 4** ($D$-similarity). *$\mathrm{DPP}(L_1)$ and $\mathrm{DPP}(L_2)$ follow the same distribution if and only if $L_1$ and $L_2$ are $D$-similar.*

Common divergences between two positive semidefinite matrices, such as the Frobenius norm and the Burg divergence, are not invariant under $D$-similarity; see Appendix **??** for further discussion.

## 4 Methodology

We address the problem of selecting a subset of prompts to maximize the reliability of a language model under a constrained inference budget. The core challenge is that naive selection strategies often choose prompts that produce correlated failures, where multiple prompts lead to similar incorrect responses or safety failures. This wastes the sampling budget and limits reliability improvements. Formally, we are given a universe of $M$ instruction prompts $\mathcal{M} = \{p_m\}_{m=1}^{M}$. For any input question $q$, our goal is to select a subset $S \subseteq \mathcal{M}$ of size $k \ll M$ that maximizes reliability for the specific task. Reliability is defined differently across application scenarios. For utility tasks (such as mathematics and code generation), we aim to maximize pass@$k$, the probability that at least one response is correct. For safety tasks, we aim to minimize false negatives, where unsafe content bypasses detection across all prompt variants. We begin with an overview of our method in Section 4.1 and then introduce the training phase in Section 4.2 and the inference phase in Section 4.3.

Figure 1: Training and inference pipelines for DPP-based prompt sampling. (a) Training (left): From each prompt–question pair, we extract hidden activations and correctness labels to construct a target, query-conditioned DPP kernel balancing quality and diversity. A lightweight encoder is optimized with a KL divergence objective (using an unbiased gradient estimator). (b) Inference (right): For a new query, the encoder yields features for all prompts. We build a Gram DPP kernel $L = F^\top F$ and sample $k$ prompts to boost accuracy (pass@$k$) or reduce false negatives for safety.

## 4.1 OVERALL OPERATIONAL PROCEDURES

Our solution consists of two phases, which is illustrated in Figure 1.

**Training Phase.** During training, we learn a query-conditioned prompt selection strategy. For each training question $q_i$ where $i \in [Q]$, and for each prompt $p_m$ where $m \in [M]$, we form the input sequence $x_{im} = [p_m, q_i]$ and extract its hidden representation $h_{im}$ from a base language model. Here, we hypothesize that the diversity in the hidden representation $h_{im}$ will lead to diversity in the outcome. Thus, we can use the kernel built from $h_{im}$ to obtain the diversity signal. We test this hypothesis in Appendix F. Subsequentaly, we train a lightweight encoder $\phi(\cdot; \theta)$ that maps $x_{im}$ to a lower-dimensional feature vector $f_{im} = \phi(x_{im}; \theta) \in \mathbb{R}^d$. For each question $q_i$, we construct the feature matrix $F_i = [f_{i1}, \ldots, f_{iM}] \in \mathbb{R}^{d \times M}$, whose Gram matrix $F_i^\top F_i$ parameterizes a Determinantal Point Process over the prompt universe. The encoder is optimized to match a target distribution that strikes a balance between prompt quality and diversity.

**Inference Phase.** At inference time, given a new question $q$ from distribution $\mathbb{Q}$, we compute for each prompt $p_m$ the feature vector $f_m = \phi([p_m, q]; \theta)$. We form the kernel matrix $L = F^\top F$ where $F = [f_1, \ldots, f_M]$, and sample a subset $S$ of size $k$ from $\mathrm{DPP}(L)$. This selected subset $S$ maximizes reliability by ensuring prompts are both high-quality and diverse in their induced model behaviors.

## 4.2 EFFECTIVE TRAINING TO APPROXIMATE TARGETED SUBSET SELECTION KERNEL

The key innovation of our approach is learning a query-conditioned DPP kernel that balances prompt quality and diversity based on the input question.

**Target Kernel Construction.** We construct a target DPP kernel $L_i$ for each training question $q_i$ that embodies the ideal trade-off between quality and diversity. For each prompt $p_m$, we obtain a hidden state $h_{im}$ from a base model and a binary accuracy label $a_{im} \in \{0, 1\}$ indicating correctness. We create a diversity-promoting kernel $\hat{L}_i$ using a Gaussian kernel between hidden states:

$$(\hat{L}_i)_{m,m'} = \exp\left(-\frac{\|h_{i,m} - h_{i,m'}\|_2^2}{2\sigma^2}\right) \qquad \forall m, m' \in \mathcal{M},$$

where $\sigma > 0$ is the width of the Gaussian kernel.

To integrate the accuracy information, we first need to convert to the marginal kernel formulation. We compute $\hat{K}_i = \hat{L}_i(I + \hat{L}_i)^{-1}$ and create a combined kernel:

$$K_i = (1 - \tau)\hat{K}_i + \tau \cdot \mathrm{diag}(a_{i1}, \ldots, a_{iM}),$$

where $\tau \in [0, 1]$ is a weighting parameter to balance the diversity and quality (accuracy) information. The diversity-quality target kernel for training question $q_i$ is $L_i = K_i(I - K_i)^{-1}$. Since the marginal kernel matrix $\hat{K}_i$ is $0 \preceq \hat{K}_i \prec I$, and $a_{im}$ takes value from 0 or 1, it follows that $0 \preceq K_i \prec I$. Consequently, $I - K_i$ is positive definite and hence invertible.

**Learning Objective.** We train the encoder $\phi$ to minimize the expected KL divergence between the target DPP distribution and the predicted distribution:

$$\min_{\phi} \mathbb{E}_{q \sim \mathbb{Q}} \left[ \mathrm{KL}\big(\mathrm{DPP}(L_q) \, \| \, \mathrm{DPP}(F_q^{\top} F_q)\big) \right], \tag{2}$$

which is invariant under D-smilarity. Basic calculation in Appendix A.1 shows that problem (2) can be written as a stochastic program

$$\min_{\phi} \mathbb{E}_{q \sim \mathbb{Q}} \left[ -\mathbb{E}_{\mathbb{P}_{L_q}}[\log \det((F_q^{\top} F_q)_{\tilde{Y}})] + \log \det(I + F_q^{\top} F_q) \right]. \tag{3}$$

**Efficient Gradient Estimation.** The objective function in problem (3) involves an expectation over the question distribution $\mathbb{Q}$ and another expectation over the target DPP distribution $\mathbb{P}_{L_q}$, both of which are intractable. We approximate the outer expectation using a mini-batch of training questions. The inner expectation, $\mathbb{E}_{\mathbb{P}_{L_q}}[\cdot]$, requires summing over all $2^M$ subsets, which is computationally infeasible. We approximate this expectation using importance sampling: Let $\mathbb{U}$ be the uniform distribution over $2^{[M]}$, which is independent of $\mathbb{Q}$. Let $\{q_i, i = 1, \ldots, Q\}$ be i.i.d. samples from the question distribution $\mathbb{Q}$. For each $i$, let $\{Y_{i,b}, b = 1, \ldots, B_i\}$ be i.i.d. samples from $\mathbb{U}$. With importance sampling, the approximated loss for (3) is

$$\frac{1}{Q} \sum_{i=1}^{Q} \left[ -\frac{2^M}{B_i} \sum_{b=1}^{B_i} \mathbb{P}_{L_i}(Y_{i,b}) \log \det((F_i^{\top} F_i)_{Y_{i,b}}) + \log \det(I + F_i^{\top} F_i) \right]. \tag{4}$$

The following Proposition 5 provides an unbiased gradient estimator for the expected KL loss minimization problem (3), facilitating stable, efficient end-to-end training with standard first-order optimizers (e.g., SGD, Adam).

**Proposition 5** (Unbiased gradient estimator of the loss function). *Assume that*

$$\mathbb{E}_{q \sim \mathbb{Q}} \left[ \sup_{\phi \in \Phi} \|\nabla_{\phi}[\log \det(I + F_q^{\top} F_q)]\| \right] < \infty, \quad \mathbb{E}_{q \sim \mathbb{Q}} \left[ \sup_{\phi \in \Phi} \|\mathbb{E}_{L_q}[\log \det((F_q^{\top} F_q)_{\tilde{Y}})]\| \right] < \infty,$$

*where $\Phi$ is a neighborhood surrounding the parameters of $\phi$. Then an unbiased estimator for the gradient of the minimization problem* (2) *is*

$$\frac{1}{Q} \sum_{i=1}^{Q} \left[ -\frac{2^M}{B_i} \sum_{b=1}^{B_i} \mathbb{P}_{L_i}(Y_{i,b}) \nabla_{\phi} \log \det((F_i^{\top} F_i)_{Y_{i,b}}) + \nabla_{\phi} \log \det(I + F_i^{\top} F_i) \right]. \tag{5}$$

**Encoder Architecture.** The encoder $\phi$ is implemented as a lightweight neural network that maps prompt-question pairs to a $d$-dimensional feature space. The network uses ReLU activations and orthogonal initialization to ensure stable training. The feature dimension $d$ is chosen to be larger than the number of prompts $M$ to ensure the Gram matrix is full-rank. We include the architecture design details in Appendix B.

### 4.3 EFFICIENT TEST TIME SUBSET SAMPLING VIA APPROXIMATED DPP KERNEL

At inference time, our method efficiently selects a diverse and high-quality subset of prompts for each new question.

**Feature Extraction.** Given a new question $q$, we form the input $[p_m, q]$ for each prompt $p_m \in \mathcal{M}$. These inputs are processed through the trained encoder $\phi$ to obtain feature vectors $f_m = \phi([p_m, q]; \theta)$. The feature matrix $F = [f_1, \ldots, f_M]$ is constructed in a single forward pass.

**Kernel Construction and Efficient Sampling.** We compute the DPP kernel matrix as $L = F^{\top} F$. This linear kernel captures both individual prompt qualities (diagonal elements) and pairwise similarities (off-diagonal elements). The inference-time computation is dominated by the forward passes through the lightweight encoder. Since the encoder has minimal parameters compared to the base language model, the additional computational cost is negligible. To further reduce complexity, we accelerate DPP sampling by employing an MAP approximation (Civril & Magdon-Ismail, 2009). The sampling process favors sets that contain high-quality prompts while maintaining diversity.

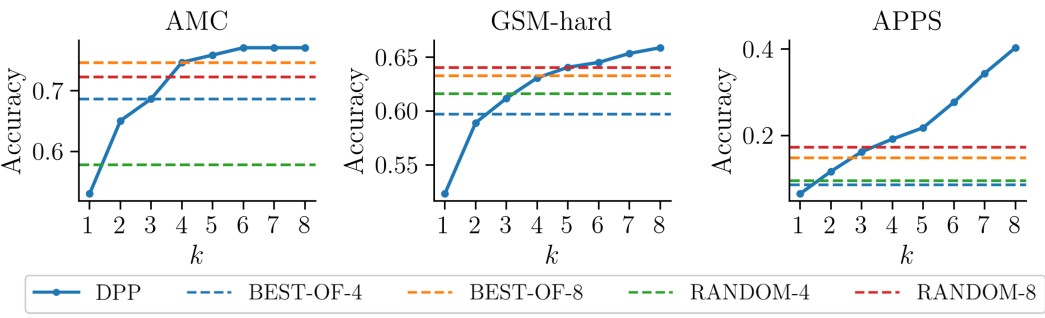

Figure 2: DPP-based prompt selection trained on math data. Scaling with prompt budget $k$. The DPP selector improves pass@$k$ more quickly than BEST-OF-K and RANDOM-K.

Table 1: Results across benchmarks and domains. We compare BEST-OF-K, RANDOM-K, and our DPP-based methods trained on different domains (MATH, APPS). Metrics are pass@$k$-style accuracy for math (MATH, AMC, AIME, GSM-hard) and code (APPS). Higher is better.

| Method | Train | MATH_test | AMC | AIME | GSM-hard | Avg | APPS_test |
|--------|-------|-----------|-----|------|----------|-----|-----------|
| | | | | **Task** | | | |
| | | BEST-OF-K | | | | | |
| BEST-OF-4 | NA | 67.75 | 68.67 | 16.67 | 59.74 | 53.21 | 8.66 |
| BEST-OF-8 | NA | 74.40 | 74.70 | 23.33 | 63.31 | 58.94 | 14.90 |
| BEST-OF-16 | NA | 78.25 | 79.52 | 26.67 | 65.43 | 62.47 | 24.60 |
| BEST-OF-24 | NA | 80.36 | 81.93 | 26.67 | 66.72 | 63.92 | 30.18 |
| | | RANDOM-K | | | | | |
| RANDOM-4 | NA | 66.69 | 57.83 | 13.33 | 61.64 | 49.87 | 9.54 |
| RANDOM-8 | NA | 72.05 | 72.29 | 23.33 | 64.06 | 57.93 | 17.32 |
| RANDOM-16 | NA | 76.36 | 81.93 | 26.67 | 66.94 | 62.98 | 29.78 |
| RANDOM-24 | NA | 78.55 | 86.75 | 26.67 | 68.39 | 65.09 | 39.86 |
| | | DPP-LAYER-15 (ours) | | | | | |
| DPP-LAYER-15 | MATH_train | 66.31 | 74.70 | 20.00 | 63.08 | 56.02 | 19.18 |
| DPP-LAYER-15 | APPS_train | 67.30 | 62.65 | 16.67 | 62.93 | 52.39 | 26.72 |

# 5 NUMERICAL EXPERIMENTS

This section examines whether query-conditioned DPP prompt selection enhances reliability within a fixed budget for generation.

## 5.1 EXPERIMENT SETUP

**Datasets.** We train and evaluate in three domains. (1) **Mathematics:** We train on the training set of MATH (Hendrycks et al., 2021b) and, unless otherwise specified, restrict supervision to Level 5 problems for higher difficulty. We evaluate our model on the test set of MATH (Hendrycks et al., 2021b) and AMC (Li et al., 2024), AIME (Li et al., 2024), and GSM-hard (Gao et al., 2023) datasets. (2) **Code generation:** We train on the training set of APPS (Hendrycks et al., 2021a) and evaluate on its testing set. (3) **Safety guardrails:** We train on a 10,000 example subset of the training set of WildGuard (Han et al., 2024) and evaluate on the testing set.

**Model backbone.** We freeze domain backbones used to obtain hidden representations $h_{im}$ and generate task outputs, and we train the lightweight encoder $\phi$ only. For generation backbones, we use QWEN2.5-MATH-1.5B-INSTRUCT (Yang et al., 2024) for math task, QWEN2.5-CODER-1.5B-INSTRUCT (Hui et al., 2024) for code generation, and QWEN2.5-1.5B-INSTRUCT (Team, 2024) for

safety guardrail. For the lightweight encoder, we use QWEN2.5-INSTRUCT-0.5B (Team, 2024). Unless stated otherwise, we extract $h_{im}$ from the frozen generation backbone.

**Training.** We optimize $\phi$ with the objective in (3), using mini-batches over questions and importance sampling over subsets as in (4). To warmup early training, we add Frobenius penalty $\eta \| L_q - F_q^\top F_q \|_F$ with a very small coefficient $\eta$. We analyze this in detail in 5.2. All generation backbones remain frozen during training.

**Evaluation protocol and metrics.** We evaluate with a fixed prompt universe $\mathcal{M}$ per domain and a budget of $k$ prompts per input. We report pass@$k$-style metrics by default and adopt task-specific correctness checks. For the math task, we compute accuracy (pass@$k$): a question is correct if any of the $k$ generations exactly matches the gold answer. For code generation, we compute accuracy (pass@$k$): a problem is correct if any of the $k$ generations pass all official unit tests. For safety guardrails, we compute recall@$k$ as the primary metric for detecting harmful content. For a selected subset $\mathcal{S}$ of size $k$, we map each prompted judgment to Yes/No. If any response in $S$ is Yes, we predict *harmful*; if *all* are No, we predict *non-harmful*. We then measure recall on harmful examples.

**Baselines and budgets.** We compare against two training-free baselines under matched budgets.

1. BEST-OF-K. We generate $k$ samples using the necessary system prompt only with temperature 0.4, and compute pass@$k$ with the task-specific checker (no prompt diversity).

2. RANDOM-K. We uniformly sample $k$ prompts from $\mathcal{M}$, generate one output per prompt, and compute the task-specific pass@$k$.

We sweep $k \in \{4, 8, 16, 24\}$ for baselines to demonstrate scaling with budget. For our DPP-based methods, we report $k = 4$ in Table 1 by default.

## 5.2 RELIABILITY RESULTS

**Math Reasoning and Code Generation.** Table 1 summarizes the accuracy for math and code generation tasks. With a budget of $k = 4$, our DPP-based prompt selection method achieves competitive or superior performance against both the BEST-OF-K and RANDOM-K baselines on most benchmarks. The learned encoder also demonstrates generalization across domains. For instance, an encoder trained on MATH improves performance on the APPS code generation task, and vice-versa, indicating that it captures a generalizable notion of prompt quality and diversity. Our method achieves this performance with high budget efficiency; on the APPS benchmark, our DPP approach with just four prompts surpasses the performance of baselines that use a significantly larger generation budget. This efficiency underscores the benefit of jointly optimizing for prompt quality and diversity. Finally, the comparison between baselines reveals that leveraging prompt diversity via RANDOM-K often yields better results than relying solely on decoding diversity with BEST-OF-K, particularly on the code generation task. This observation supports our premise that for smaller backbones, strategically selecting for input-side prompt diversity is more effective than relying on decode-side diversity alone.

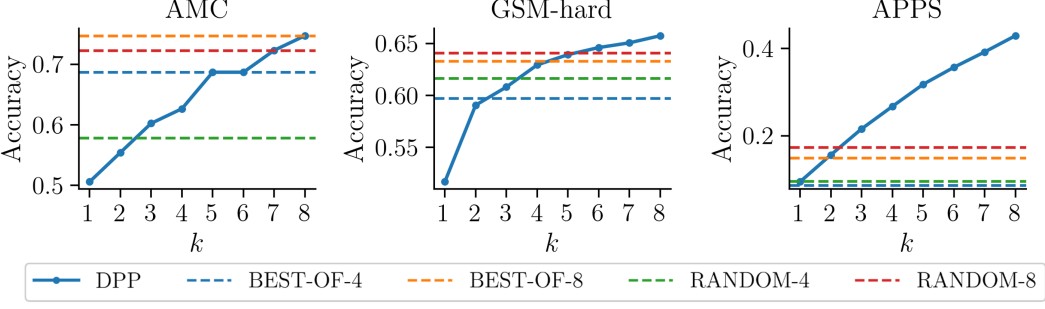

Figure 3: DPP-based prompt selection trained on code data. Scaling with prompt budget $k$. The DPP selector improves pass@$k$ more quickly than BEST-OF-K and RANDOM-K.

**Safety Guardrail.** Figure 4 shows that for safety guardrail tasks, our method consistently improves recall@$k$ over both RANDOM-K and BEST-OF-K baselines across all budget levels. Notably, our method's recall scales efficiently as the budget increases, demonstrating its ability to effectively utilize each additional prompt for improved reliability. In contrast, the baseline methods scale less effectively with the additional budget, suggesting they suffer from correlated failures where different prompts fail to detect the same harmful content. This result validates our hypothesis that query-agnostic selection strategies cannot reliably leverage prompt diversity to make robust safety judgments.

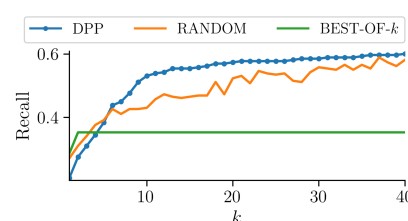

Figure 4: Safety guardrail scaling on WildGuard test.

**Ablation Study.** We analyze three key factors to understand the robustness of our method: the impact of the prompt budget $k$, the encoder's ability to generalize across domains, and the design of the training objective.

First, we examine performance scaling with the prompt budget $k$. As Figures 2, 3, and 4 illustrate, our method's pass@$k$ and recall@$k$ gains generally meet or exceed those of the baselines. The benefits are particularly pronounced at smaller budgets, demonstrating high efficiency. For instance, on the APPS benchmark, our method with $k = 4$ already outperforms BEST-OF-16. This efficiency is crucial in resource-constrained scenarios, as it enables significant reliability improvements without a corresponding increase in computational cost.

Second, we assess the cross-domain generalization of the learned encoder by training on one domain and evaluating on another (MATH and APPS). In both transfer settings, our method maintains a competitive edge against the baselines, showcasing its adaptability. Although the gains are smaller than when training in-domain, these results confirm that our query-conditioned kernel learns generalizable principles of prompt quality and diversity. This adaptability makes our approach effective for new tasks without requiring task-specific encoder training.

Third, we analyze the training objective for DPP kernel learning from both theoretical and empirical perspectives. Empirically, during the training phase, we use a small Frobenius norm penalty term to warm up the optimization process. Theoretically, however, this Frobenius norm term can be unstable, especially when the kernel matrix has small eigenvalues, which can lead to training instability. We formalize this analysis in Appendix A.2. Empirically, we observe that including a small Frobenius norm term early in training can help accelerate convergence. However, in later training stages, the learning process depends primarily on our developed KL-divergence gradient estimator, which provides more stable and effective updates.

## 6 CONCLUSIONS

We present a query-conditioned prompt selection method based on Determinantal Point Processes. The method learns a kernel that combines per-prompt quality and pairwise diversity signals derived from hidden activations, trains a small encoder with a Kullback–Leibler objective using an unbiased gradient estimator that is invariant under D-similarity, and selects $k$ prompts with a fast greedy MAP procedure. Across mathematics, code, and safety guardrail tasks, the approach increases pass@$k$ and recall@$k$ relative to BEST-OF-K and RANDOM-K while adding little inference cost. The gains are largest at small budgets, which is important in settings where each extra generation is costly. The method also transfers across domains, which indicates that the encoder learns general selection cues that are not tied to a single dataset. This work shows that structured input diversity reduces correlated failures more effectively than decode-side sampling alone on small and medium backbones. By shifting computation to a lightweight encoder and by sampling from a learned DPP at test time, the system efficiently improves reliability without changing the base model weights.

**Ethics statement.** We evaluate on public benchmarks for safety guardrails. Our models process this content solely for the purpose of detecting harmful content. However, a method that selects prompts that change model behavior could be used to search for prompts that weaken guardrails. To mitigate this risk, we limit the candidate prompts to instruction-style prompts that require detection solutions.

**Reproducibility statement.** We release all hyperparameters in Appendix D. We release the prompt libraries used in each domain in Appendix E. We include the computation details in Appendix D.

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

## A  TECHNICAL SUPPORT

### A.1  CALCULATION OF THE KULLBACK-LEIBLER DIVERGENCE

Basic calculation gives us

$$\mathrm{KL}(\mathrm{DPP}(L_q)\|\mathrm{DPP}(F_q^\top F_q))$$

$$= \sum_{Y\subseteq[M]} \mathbb{P}_{L_q}(Y) \log\Big(\frac{\det(L_{q,Y})}{\det(I+L_q)} \times \frac{\det(I+F_q^\top F_q)}{\det((F_q^\top F_q)_Y)}\Big)$$

$$= \sum_{Y\subseteq[M]} \mathbb{P}_{L_q}(Y) \log\frac{\det(L_{q,Y})}{\det((F_q^\top F_q)_Y)} + \log\det(I+F_q^\top F_q) - \log\det(I+L_q)$$

$$= \mathbb{E}_{\mathbb{P}_{L_q}}[\log\det(L_{q,\tilde{Y}})] - \mathbb{E}_{\mathbb{P}_{L_q}}[\log\det((F_q^\top F_q)_{\tilde{Y}})] + \log\det(I+F_q^\top F_q) - \log\det(I+L_q)$$

$$= -\mathbb{E}_{\mathbb{P}_{L_q}}[\log\det((F_q^\top F_q)_{\tilde{Y}})] + \log\det(I+F_q^\top F_q) + C,$$

where $C_q = \mathbb{E}_{\mathbb{P}_{L_q}}[\log\det(L_{q,\tilde{Y}})] - \log\det(I+L_q)$ is a constant independent of $F_q$.

### A.2  ADDITIONAL EXAMPLES: CHALLENGES IN LEARNING DPPS

Consider the Burg divergence between $L_1$ and $L_2 \in \mathbb{R}^{M\times M}$, which is formally defined as

$$\mathrm{Burg}(L_1 \| L_2) = -\mathrm{Tr}[L_1 L_2^{-1}] + \log\det(L_1 L_2^{-1}) - M.$$

Consider the kernel matrix $L_1$ and the scaling $D$:

$$L_1 = \begin{bmatrix} 1 & 0.5 \\ 0.5 & 1 \end{bmatrix}, \ D = \begin{bmatrix} 1 & 0 \\ 0 & -1 \end{bmatrix}.$$

If we set $L_2 = DL_1D$, then by Theorem 4, $\mathrm{DPP}(L_1)$ and $\mathrm{DPP}(L_2)$ are probabilistically equivalent. Nevertheless, by replacing the numerical values of $L_1$ and $L_2$, we find $\mathrm{Burg}(L_1 \| L_2) = -16/3$. This implies that two matrices that induce probabilistically the same DPP could have a non-zero Burg measurement. In fact, if we use *any* distance or divergence in the space of positive (semi)definite matrices, we obtain a strictly positive value when we measure the difference between $L_1$ and $L_2$. The problem becomes more pronounced if we use the Frobenius distance to measure the dissimilarity between matrices: two matrices could have an arbitrarily small Frobenius distance, but the corresponding DPPs are fundamentally different. Consider

$$L = \begin{bmatrix} 1 & 1 \\ 1 & 1 \end{bmatrix}, \quad L_n = \begin{bmatrix} 1+\frac{1}{n} & 1 \\ 1 & 1 \end{bmatrix}$$

for $n \geq 1$. Since $\det(L) = 0$, we have the following:

$$\mathbb{P}_L(\{1,2\}) = \frac{\det L}{\det(I+L)} = 0,$$

which means that $\mathrm{DPP}(L)$ never selects both items. While for $L_n$,

$$\mathbb{P}_{L_n}(\{1,2\}) = \frac{\det L_n}{\det(I+L_n)} = \frac{1}{3n-2},$$

which means that for any $n \geq 1$, $\mathrm{DPP}(L_n)$ always selects both items with some positive probability. However, we can verify $\lim_{n\to\infty}\|L_n - L\|_F = 0$. This example implies that a small perturbation of the kernel $L$, measured by the Frobenius norm, can result in a qualitative collapse of the model's possibilities.

We further provide a result showing that the KL divergence between the two DPPs is upper-bounded by a constant times the Frobenius norm distance between the kernels. Noting that a tiny eigenvalue $\mu_1$ can blow up the prefactor and make the bound unstable. Consequently, using the Frobenius gap directly as a loss can be unstable for poorly conditioned kernels, as small eigenvalues can magnify its effect on the KL divergence.

**Proposition 6.** *Let $L_1, L_2 \in \mathbb{R}^{M\times M}$ be two positive definite matrices with eigenvalues $\{\lambda_i\}_{i=1}^M$ and $\{\mu_i\}_{i=1}^M$, respectively. Assume that $\{\lambda_i\}_{i=1}^M$ and $\{\mu_i\}_{i=1}^M$ are sorted in ascending order. Then, we have*

$$\mathrm{KL}(\mathrm{DPP}(L_1)\|\mathrm{DPP}(L_2)) \leq \sqrt{M}(\frac{1}{1+\lambda_1} + \frac{1}{\mu_1})\|L_1 - L_2\|_F. \tag{6}$$

### A.3 PROOFS

*Proof of Proposition 5.* We first recite the two conditions in the statement,

$$\mathbb{E}_{q\sim\mathbb{Q}}\left[\sup_{\phi\in\Phi}\|\nabla_\phi\left[\log\det(I+F_q^\top F_q)\right]\|\right]<\infty, \tag{7}$$

and

$$\mathbb{E}_{q\sim\mathbb{Q}}\left[\sup_{\phi\in\Phi}\|\mathbb{E}_{L_q}[\log\det((F_q^\top F_q)_{\tilde{Y}})]\|\right]<\infty. \tag{8}$$

Taking expectation on (5) gives us

$$\mathbb{E}_{\{q_i\},\{Y_{i,b}\}}\left[\frac{1}{Q}\sum_{i=1}^{Q}\left[-\frac{2^M}{B_i}\sum_{b=1}^{B_i}\mathbb{P}_{L_i}(Y_{i,b})\nabla_\phi\log\det((F_i^\top F_i)_{Y_{i,b}})+\nabla_\phi\log\det(I+F_i^\top F_i)\right]\right]$$

$$=\frac{1}{Q}\sum_{i=1}^{Q}\mathbb{E}_{q_i,\{Y_{i,b}\}}\left[-\frac{2^M}{B_i}\sum_{b=1}^{B_i}\mathbb{P}_{L_i}(Y_{i,b})\nabla_\phi\log\det((F_i^\top F_i)_{Y_{i,b}})+\nabla_\phi\log\det(I+F_i^\top F_i)\right]$$

$$=\frac{1}{Q}\sum_{i=1}^{Q}\mathbb{E}_{q_i,\{Y_{i,b}\}}\left[-\frac{2^M}{B_i}\sum_{b=1}^{B_i}\mathbb{P}_{L_i}(Y_{i,b})\nabla_\phi\log\det((F_i^\top F_i)_{Y_{i,b}})\right]$$

$$+\frac{1}{Q}\sum_{i=1}^{Q}\mathbb{E}_{q_i}\left[\nabla_\phi\log\det(I+F_i^\top F_i)\right]. \tag{9}$$

For the first term in (9), we have

$$\frac{1}{Q}\sum_{i=1}^{Q}\mathbb{E}_{q_i,\{Y_{i,b}\}}\left[-\frac{2^M}{B_i}\sum_{b=1}^{B_i}\mathbb{P}_{L_i}(Y_{i,b})\nabla_\phi\log\det((F_i^\top F_i)_{Y_{i,b}})\right]$$

$$=\frac{1}{Q}\sum_{i=1}^{Q}\mathbb{E}_{q_i}\left[-\frac{1}{B_i}\sum_{b=1}^{B_i}\mathbb{E}_{Y_{i,b}}\left[\frac{\mathbb{P}_{L_i}(Y_{i,b})}{2^{-M}}\nabla_\phi\log\det((F_i^\top F_i)_{Y_{i,b}})\mid q_i\right]\right]$$

$$=\frac{1}{Q}\sum_{i=1}^{Q}\mathbb{E}_{q_i}\left[-\frac{1}{B_i}\sum_{b=1}^{B_i}\nabla_\phi\mathbb{E}_{Y_{i,b}}\left[\frac{\mathbb{P}_{L_i}(Y_{i,b})}{2^{-M}}\log\det((F_i^\top F_i)_{Y_{i,b}})\mid q_i\right]\right] \qquad (Y_{i,b}\sim\mathbb{U}\text{ is a finite distribution})$$

$$=\frac{1}{Q}\sum_{i=1}^{Q}\mathbb{E}_{q_i}\left[-\frac{1}{B_i}\sum_{b=1}^{B_i}\nabla_\phi\sum_{Y_{i,b}\in2^{[M]}}\mathbb{P}_{L_i}(Y_{i,b})\log\det((F_i^\top F_i)_{Y_{i,b}})\right] \qquad (Y_{i,b}\sim\mathbb{U}\text{ is uniform over }2^{[M]})$$

$$=\frac{1}{Q}\sum_{i=1}^{Q}\mathbb{E}_{q_i}\left[\frac{1}{B_i}\sum_{b=1}^{B_i}\nabla_\phi\mathbb{E}_{L_i}\left[-\log\det((F_i^\top F_i)_{\tilde{Y}})\right]\right]$$

$$=\frac{1}{Q}\sum_{i=1}^{Q}\mathbb{E}_{q_i}\left[\nabla_\phi\mathbb{E}_{L_i}\left[-\log\det((F_i^\top F_i)_{\tilde{Y}})\right]\right]$$

$$=\frac{1}{Q}\sum_{i=1}^{Q}\nabla_\phi\mathbb{E}_{q_i}\left[\mathbb{E}_{L_i}\left[-\log\det((F_i^\top F_i)_{\tilde{Y}})\right]\right] \qquad ((8)\text{ and dominated convergence theorem})$$

$$=\nabla_\phi\mathbb{E}_q\left[\mathbb{E}_{L_q}\left[-\log\det((F_q^\top F_q)_{\tilde{Y}})\right]\right]. \tag{10}$$

Condition (7) together with the dominated convergence theorem gives us that

$$\frac{1}{Q}\sum_{i=1}^{Q}\mathbb{E}_{q_i}\left[\nabla_\phi\log\det(I+F_i^\top F_i)\right]=\frac{1}{Q}\sum_{i=1}^{Q}\nabla_\phi\mathbb{E}_{q_i}\left[\log\det(I+F_i^\top F_i)\right]$$

$$=\nabla_\phi\mathbb{E}_q\left[\log\det(I+F_q^\top F_q)\right]. \tag{11}$$

Substituting (10) and (11) into (9), then we have

$$\mathbb{E}_{\{q_i\},\{Y_{i,b}\}}\left[\frac{1}{Q}\sum_{i=1}^{Q}\left[-\frac{2^M}{B_i}\sum_{b=1}^{B_i}\mathbb{P}_{L_i}(Y_{i,b})\nabla_\phi\log\det((F_i^\top F_i)_{Y_{i,b}})+\nabla_\phi\log\det(I+F_i^\top F_i)\right]\right]$$

$$=\nabla_\phi\mathbb{E}_q\left[\mathbb{E}_{L_q}\left[-\log\det((F_q^\top F_q)_{\tilde{Y}})\right]\right]+\nabla_\phi\mathbb{E}_q\left[\log\det(I+F_q^\top F_q)\right]$$

$$=\nabla_\phi\mathbb{E}_q\left[\mathbb{E}_{L_q}\left[-\log\det((F_q^\top F_q)_{\tilde{Y}})\right]+\log\det(I+F_q^\top F_q)\right],$$

which proves the unbiasedness of (5). □

*Proof of Proposition 6.* By definition of KL divergence, we have

$$\mathrm{KL}(\mathrm{DPP}(L_1)\|\mathrm{DPP}(L_2))=\sum_Y\mathbb{P}_{L_1}(Y)\log\frac{\det(L_{1,Y})}{\det(L_{2,Y})}+\log\det(I+L_2)-\log\det(I+L_1).$$

Firstly we bound the term $\log\det(I+L_2)-\log\det(I+L_1)$. Since $X\mapsto\log\det(X)$ is concave and $\nabla_X\log\det(X)=(X^{-1})^\top$, it holds that

$$\log\det(I+L_2)-\log\det(I+L_1)\le\langle(I+L_1)^{-1},L_2-L_1\rangle$$
$$\le\|(I+L_1)^{-1}\|_F\|L_2-L_1\|_F,$$

where the last inequality comes from the Cauchy-Schwarz inequality. For $\|(I+L_1)^{-1}\|_F$, we can bound

$$\|(I+L_1)^{-1}\|_F^2=\sum_{i=1}^{M}\frac{1}{(1+\lambda_i)^2}\le\frac{M}{(1+\lambda_1)^2}\qquad\Longrightarrow\qquad\|(I+L_1)^{-1}\|_F\le\frac{\sqrt{M}}{1+\lambda_1}.$$

Thus, we have

$$\log\det(I+L_2)-\log\det(I+L_1)\le\|(I+L_1)^{-1}\|_F\|L_2-L_1\|_F\le\frac{\sqrt{M}}{1+\lambda_1}\|L_2-L_1\|_F. \quad(12)$$

Next, we bound the term $\sum_Y\mathcal{P}_{L_1}(Y)\log\frac{\det(L_{1,Y})}{\det(L_{2,Y})}$. Again with the concavity of $\log\det(X)$, we have

$$\log\frac{\det(L_{1,Y})}{\det(L_{2,Y})}=\log\det(L_{1,Y})-\log\det(L_{2,Y})$$
$$\le\langle(L_{2,Y}^{-1})^\top,L_{1,Y}-L_{2,Y}\rangle$$
$$\le\|L_{2,Y}^{-1}\|_F\|L_{1,Y}-L_{2,Y}\|_F.$$

For $\|L_{2,Y}^{-1}\|_F$, we have

$$\|L_{2,Y}^{-1}\|_F\le\sqrt{M}\|L_{2,Y}^{-1}\|_2=\sqrt{M}\lambda_{max}(L_{2,Y}^{-1})=\frac{\sqrt{M}}{\lambda_{min}(L_{2,Y})}\le\frac{\sqrt{M}}{\mu_1}, \quad(13)$$

where the first inequality comes from the fact that $\|A\|_F\le\sqrt{M}\|A\|_2$ for a matrix $A$ with rank $M$; the last inequality comes from the Cauchy interlacing theorem. For $\|L_{1,Y}-L_{2,Y}\|_F$, since $L_{1,Y}-L_{2,Y}$ is a submatrix of $L_1-L_2$,

$$\|L_{1,Y}-L_{2,Y}\|_F=\|(L_1-L_2)_Y\|_F\le\|L_1-L_2\|_F.$$

Thus, we have

$$\log\frac{\det(L_{1,Y})}{\det(L_{2,Y})}\le\|L_{2,Y}^{-1}\|_F\|L_{1,Y}-L_{2,Y}\|_F$$
$$\le\frac{\sqrt{M}}{\sqrt{\mu_1}}\|L_1-L_2\|_F. \quad(14)$$

Combining (12) and (14) together gives us

$$\mathrm{KL}(\mathrm{DPP}(L_1)\|\mathrm{DPP}(L_2)) = \sum_Y \mathcal{P}_{L_1}(Y) \log \frac{\det(L_{1,Y})}{\det(L_{2,Y})} + \log \det(I + L_2) - \log \det(I + L_1)$$

$$\leq \frac{\sqrt{M}}{1 + \lambda_1}\|L_1 - L_2\|_F + \frac{\sqrt{M}}{\mu_1}\|L_1 - L_2\|_F$$

$$= \sqrt{M}(\frac{1}{1 + \lambda_1} + \frac{1}{\mu_1})\|L_1 - L_2\|_F,$$

which completes the proof. □

## B  MODEL ARCHITECTURE DETAIL

- Note: the mapping $\phi$ has multiple effects:
  - The kernel on the hidden space could be a nonlinear kernel. For example, $L_i$ can be constructed using the Gaussian kernel. However, after passing inputs through the mapping $\phi$, we will use a linear kernel
  - Dimensionality reduction: $d$ could be smaller than the dimension of the hidden vector
- Initialization to ensure a non-degenerate matrix
  - Use orthogonal initialization
  - Ensure $d > M$

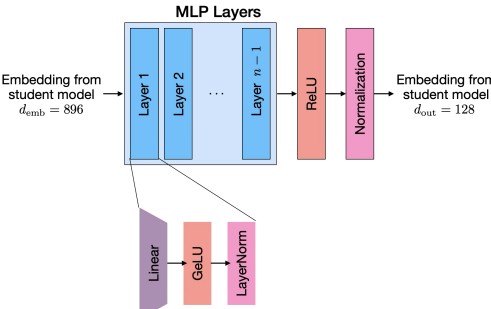

Figure 5: Lightweight encoder architecture. A small ReLU-MLP with orthogonal initialization maps each input $[p_m, q]$ to a feature $f_m \in \mathbb{R}^d$. ReLU promotes nonnegativity for identifiability, and choosing $d > M$ helps avoid degeneracy.

## C  USAGE OF LARGE LANGUAGE MODELS

We disclose that a general-purpose large language model (LLM) was used solely as a writing assistant to polish the manuscript's prose. It did not contribute any novel ideas or technical content. All factual claims, numbers, and citations were verified by the authors, who take full responsibility for the manuscript's content. The LLM is not an author and is not eligible for authorship

## D  IMPLEMENTATION HYPERPARAMETERS

This appendix lists all hyperparameters used in training and inference. We keep the same settings across domains unless stated. All training runs execute on NVIDIA A5000 GPUs with 24 GB memory.

Table 2: Hyperparameters for training and inference.

| Name | Value |
|---|---|
| Feature dimension $d$ | 128 |
| Gaussian width $\sigma$ | $\sqrt{\text{median}_{i<j} \|h_i - h_j\|_2^2/10}$ (per question) |
| Mixing coefficient $\tau$ | 0.1 |
| Learning rate | 0.001 |
| Batch size (questions) | 576 for MATH_train; 500 for APPS_train |
| Number of training steps | 400 for MATH_train; 1100 for APPS_train |
| Importance sampling budget (subsets per question) | 1024 |
| Warm-up penalty weight $\eta$ | 0.001 |
| DPP selection mode | Greedy MAP (default) |

## E  PROMPT UNIVERSES

This appendix contains the complete set of prompts used in our experiments across different domains: mathematical reasoning, code generation, and safety guardrails.

### E.1  MATHEMATICAL REASONING PROMPTS

This collection of 80 prompts is designed to elicit diverse mathematical reasoning strategies from language models. Each prompt encourages a different approach to problem-solving, including algebraic manipulation, geometric reasoning, case analysis, and iterative methods.

```
[
    "Represent all quantities in the scenario using symbols, form
        equations that capture their relationships, manipulate
        them to isolate the target variable, calculate its value,
        then reinsert it into the original equations to ensure
        consistency with all given data.",
    "Convert the described situation into a system of symbolic
        expressions, apply suitable algebraic transformations to
        solve for the unknown, compute its numeric value, and
        validate by checking it satisfies all original conditions
        and units.",
    "Assign letters to each relevant quantity, express all
        relationships as equations, perform the necessary
        rearrangements to obtain the unknown explicitly, evaluate
        it numerically, and verify by substituting back and
        checking plausibility.",
    % more math prompts not shown
]
```

### E.2  CODE GENERATION PROMPTS

This collection of 80 prompts guides language models to generate code using different programming paradigms, styles, and optimization strategies. The prompts vary in their emphasis on functional vs. imperative approaches, memory efficiency, readability, and error handling.

```
[
    "Write a pure, functional-style solution that transforms
        inputs into the required output using small composable
        functions and expressions; avoid shared state and mutate
        nothing in place.",
```

```
        "Produce an imperative, single-pass routine that updates
            local variables as you scan the input once, using clear
            variable names and straightforward control flow.",
        "Implement a recursive solution that delegates work to itself
            on progressively smaller subproblems; ensure a well-
            defined base case and return the final value.",
        % more code prompts not shown
    ]
```

### E.3 SAFETY GUARDRAIL PROMPTS

This collection of 80 prompts is designed for toxicity detection and content moderation tasks. The prompts employ different detection strategies, sensitivity thresholds, and reasoning approaches to identify harmful content across various dimensions.

```
    [
        "Read one paragraph and reason privately. If it contains any
            of: direct insults at a person or group, dehumanization (
            comparing people to animals/objects or denying humanity),
            threats or wishes of harm, slurs/epithets, or commands to
            exclude/silence someone, output Yes; general profanity
            without a target still counts as toxic here; uncertainty
            breaks toward Yes. Output exactly one word: Yes or No.",
        "Analyze internally. Mark Yes only if the paragraph has
            targeted abuse (insults directed at someone),
            dehumanization, slurs, or explicit/implicit threats;
            general profanity or rude tone without a target is No;
            borderline cases break toward No. Output exactly one word:
             Yes or No.",
        "Think privately and score: 2 points for threats or slurs; 1
            point each for targeted insults, dehumanization, or
            harassment commands. General profanity = 0 unless targeted
            . If total greater than 2, output Yes; else No. Output
            exactly one word.",
        % more safety prompts not shown
    ]
```

### E.4 PROMPT STATISTICS

Table 3: Summary of prompt universe characteristics

| Domain | Count | Avg. Length (chars) |
|---|---|---|
| Mathematical Reasoning | 100 | 245 |
| Code Generation | 100 | 198 |
| Safety Guardrails | 100 | 312 |

### E.5 USAGE NOTES

The prompt universes are used as follows in our methodology:

- **Training Phase**: During training, we sample from the entire prompt universe to construct the target DPP kernel that balances accuracy and diversity.

- **Inference Phase**: At test time, our lightweight encoder selects a diverse subset of $k$ prompts from the full universe of $M$ prompts ($M = 100$ for each domain).

- **Domain Specificity**: Each domain uses its dedicated prompt universe, ensuring that the prompts are tailored to the specific task requirements.

The prompts are designed to cover a wide spectrum of reasoning strategies and detection approaches, enabling our DPP-based method to select complementary prompts that maximize coverage of different solution pathways or failure modes.

# F  HYPOTHESIS TESTING

In Section 4.1, we postulate that the diversity in the hidden vectors $h_{im}$ will induce diversity in the final answers. We now propose a statistical test to validate this hypothesis.

Remind that for each training question $q_i$ where $i \in [Q]$, and for each prompt $p_m$ where $m \in [M]$, we form the input sequence $[p_m, q_i]$ and extract its hidden representation $h_{im}$ from a base language model. The language model's outcome for question $i$ under prompt $m$ is a single Bernoulli realization $a_{im} \in \{0, 1\}$, where $a_{im} = 1$ indicates a successful response (correct) and $a_{im} = 0$ a failure. For any two prompts $m$ and $m'$, we form

$$Z_{i,mm'} = \|h_{im} - h_{im'}\|_2, \quad \text{and} \quad Y_{i,mm'} = |a_{im} - a_{im'}|.$$

Thus, $Z$ is the predictor representing the difference of the hidden activations, and $Y$ is the target representing the difference in the answer accuracy.

Because each question may have different bias, we consider the following fixed effect logistic regression model (Chamberlain, 1980)

$$\Pr(Y_{i,mm'} = 1 \mid Z_{i,mm'}, \alpha_i) = \frac{\exp(\alpha_i + \beta Z_{i,mm'})}{1 + \exp(\alpha_i + \beta Z_{i,mm'})} \qquad \forall(i, m, m'), \tag{15}$$

where $\alpha_i \in \mathbb{R}$ are question-specific intercepts and $\beta \in \mathbb{R}$ is the common slope parameter.

We are interested in testing whether the predictor $Z$ is positively associated with the outcome $Y$:

$$\mathcal{H}_0 : \beta \leq 0 \qquad \text{versus} \qquad \mathcal{H}_A : \beta > 0.$$

To conduct this test, we follow the procedure from McCullagh & Nelder (1989): we first solve the pooled log-likelihood maximization problem

$$\max_{\alpha_1, \ldots, \alpha_Q, \beta} \sum_{i=1}^{Q} \sum_{\substack{m, m'=1 \\ m \neq m'}}^{M} \Big[ Y_{i,mm'}(\alpha_i + \beta Z_{i,mm'}) - \log\big(1 + e^{\alpha_i + \beta Z_{i,mm'}}\big) \Big],$$

which produces the maximum likelihood estimator (MLE) $\hat{\beta}$ of the slope, together with estimates of nuisance parameters $\hat{\alpha}_i$.

Subsequently, we use the Wald test statistic for $H_0$, which is given by

$$z_{\text{Wald}} = \frac{\hat{\beta}}{\widehat{\text{SE}}(\hat{\beta})},$$

where $\widehat{\text{SE}}(\hat{\beta})$ is a standard error estimator for $\hat{\beta}$. To account for potential within-group dependence and heterogeneity, we compute a cluster-robust (sandwich) variance estimator, clustering at the question level.

We conduct the hypothesis tests on two datasets: MATH_train and APPS_train. Due to the large number of questions and prompts in each dataset, we subsample a smaller dataset as follows: First, we randomly selected 300 questions from the full set. Within each selected question, we then applied stratified sampling with respect to the binary outcome variable ($Y$) to retain 30 prompt-pairs, so that both outcome classes were proportionally preserved. Table 4 summarizes the test outcomes for the significant level of 5%.

For the Math dataset, the p-value is below the conventional 5% significance threshold, we reject the null hypothesis in favor of the alternative. The effect size, however, is small, and the test only marginally rejects the null, suggesting weak but statistically significant evidence of a positive association. For the APPS dataset, the Wald statistic is very large, yielding a one-sided p-value effectively

Table 4: One-sided Wald test of $\mathcal{H}_0 : \beta \leq 0$ vs. $\mathcal{H}_A : \beta > 0$.

| Dataset | $\widehat{\beta}$ | SE($\widehat{\beta}$) | $z_{\text{Wald}}$ | $p_{\text{one-sided}}$ | Reject Null Hypothesis? |
|---|---|---|---|---|---|
| Math dataset | 0.019588 | 0.011363 | 1.7239 | 0.0423671 | YES |
| APPS dataset | 0.339558 | 0.040726 | 8.3377 | 0 | YES |

equal to zero. This provides extremely strong evidence against the null hypothesis and in favor of a positive effect. Here, both the magnitude of the coefficient and the strength of the statistical evidence point to a robust positive association between the predictor $Z$ and the outcome $Y$. Overall, both datasets support the conclusion that the distance between hidden activation vectors is positively correlated with the difference in accuracy of the generated answers, although the strength of the evidence differs substantially: weak but significant in the Math dataset and very strong in the APPS dataset.

