# OpenReview forum: "Scaling Language Model Reliability via Determinantal Point Process Prompt Sampling"
_ICLR.cc/2026/Conference — Submitted to ICLR 2026_

### Official Review · Reviewer_GiWc · 2025-10-30

**Soundness:** 3
**Presentation:** 3
**Contribution:** 2
**Rating:** 4
**Confidence:** 4

**Summary:**

This paper proposes a determinantal point process based framework for test-time prompt selection to improve language model reliability without modifying model parameters.
A query-conditioned encoder is trained to select a subset of prompts that balance accuracy and reasoning diversity.
They suggest a training objective and an unbiased gradient estimator to train the model stably.
Experiments on mathematical reasoning, code generation, and safety guardrail tasks show consistent gains over Best-of-k and Random-k baselines, particularly under small generation budgets.

**Strengths:**

## **Strengths**

1. The paper is well written and easy to follow, with clear motivation, structure, and explanations of both the method and experiments.
2. Using Determinantal Point Processes (DPPs) as a prompt sampling distribution is a novel and interesting idea that provides a principled way to encourage both accuracy and diversity in prompt selection.
3. The design of a kernel that explicitly balances accuracy and diversity is a thoughtful and effective approach for improving reliability under limited generation budgets.
4. Emphasizing the applicability of the method to safety-related tasks, such as harmful content detection, adds practical value and broadens the potential impact of the work.

**Weaknesses:**

## **Weaknesses and Questions**

1. **Generalization to New Tasks**
   The method depends on a fixed prompt universe $(M \approx 80)$ defined before inference.
   While the encoder adapts selection to each query, it cannot easily handle unseen tasks or domains where the style or structure of valid prompts largely differs from the training dataset. Although authors provide some ablation results on cross-domain experiments, this limits how broadly the approach can generalize beyond the provided prompt pool.

2. **Compute Efficiency and “Lightweight” Definition**
   The encoder is referred to as “lightweight,” but it is implemented with **Qwen2.5-Instruct-0.5B**, which is relatively large.
   Because the encoder must process all $M$ prompts for each query, the total inference cost scales linearly with $M$.
   In the experimental setup $(M \approx 80, k \le 24)$, the encoder’s total cost could rival or exceed that of $k$ LM generations.
   The paper describes “lightweight” only in terms of parameter count, without measuring actual compute time, FLOPs, or latency.

3. **Scalability with Respect to M**
   Building and sampling from the full Gram matrix $L = F^{\top}F$ introduces $O(M^2)–O(M^3)$ complexity.
   As (M) increases, both runtime and memory overhead can become substantial.
   The paper does not include analyses showing how performance or efficiency change with larger prompt pools.

4. **Training Stability of the Kernel Matching Objective**
   The encoder is trained to match a target DPP kernel via a KL-divergence objective between two $(M \times M)$ matrices.
   Although the authors propose an unbiased gradient estimator, such optimization can be sensitive to ill-conditioned kernels and may be difficult to stabilize for complex tasks.
   The paper would benefit from more discussion or diagnostics on convergence behavior.

5. **Experimental Scope and Missing Ablations**
   Experiments focus on small-to-medium settings $(k \le 24, M \approx 80)$ and do not include analyses of larger-scale scenarios or varying encoder sizes.
   Additional studies on computation–performance trade-offs, sensitivity to $M$ (both increasing or decreasing), or efficiency under different model scales would strengthen the practical claims of scalability.

6. **Manual Construction of Instruction Prompts**
   The method relies on a manually defined set of instruction prompts, which must be prepared in advance.
   This dependence on human-curated prompts can limit the diversity and coverage of reasoning behaviors the model can explore.
   In real-world scenarios, building and maintaining such a prompt library requires domain expertise and substantial manual effort, making it difficult to ensure truly diverse or adaptive outputs without some form of automatic prompt generation or refinement.

7. **Lack of Ablation on the Mixing Coefficient (τ)**
   The paper introduces a mixing coefficient $\tau$ to balance the accuracy and diversity terms when constructing the target kernel.
   However, the experiments fix $\tau = 0.1$ without providing any ablation or sensitivity analysis.
   Since $\tau$ directly governs the trade-off between selecting accurate prompts and ensuring diversity, understanding its influence is crucial for interpreting the method’s robustness and general applicability.
   It would be helpful to include results or discussion showing how varying $\tau$ (e.g., 0.05, 0.2, 0.5) affects performance across tasks.

**Questions:**

See the above section.
Overall, I personally find the proposed approach interesting and well motivated. If the authors provide convincing answers or additional analyses addressing the weaknesses and questions I raised, I would be willing to increase my score.

---

### Official Review · Reviewer_A6pk · 2025-10-30

**Soundness:** 2
**Presentation:** 2
**Contribution:** 2
**Rating:** 2
**Confidence:** 4

**Summary:**

In the paper, the authors target the problem of increasing reliability of LLMs during inference (i.e., pass@k performance) by selecting a "diverse" yet well performing set of prompts in a query dependent way.
To do so a small encoder to parameterize a DPP kernel is trained with the kernel balancing prompt quality (inferred from accuracy labels) and diversity (based on hidden activation similarity).
At test time, the encoder selects k diverse prompts per query. One output per prompt is generated with the goal to increase pass@k performance.
To train the encoder via KL divergence, an unbiased gradient estimator is derived which relies on importance sampling.
The DPP method is evaluated on  math (MATH, AMC, AIME, GSM-hard) and code (APPS) tasks and WildGuard as a safety guardrail task.
Results show that DPP usually meets the bar of simple baselines (BEST-OF-k and RANDOM-k.) but on APPS shows strong performance.

**Strengths:**

Query-dependent prompt selection is an important and relevant topic.

Using DPPs to jointly optimize quality and encourage diversity of the set of prompts is theoretically grounded and motivated.

The unbiased gradient estimator for the KL objective via importance sampling is some technical contribution.

Testing is performed across different domains math (MATH, AMC, AIME, GSM-hard), code (APPS) and safety (WildGuard)

The authors show that encoders trained on MATH improve APPS performance which suggests some generalization.

**Weaknesses:**

The main motivation of the paper is never validated.
The central claim that baselines suffer from correlated failures (especially with small LLMs) is never demonstrated empirically or backed up by referencing existing works (not sure if such exist).

Baselines are simple do not make use of training data which results in an apple to bananas comparison.
Existing work such as TEMPERA (https://arxiv.org/abs/2211.11890) would be a better competitor (and TEMPERA also questions the general usefulness of having more than k = 1 per prompt in a query dependent manner).
Maybe another baseline also could be a simple learned prompt ranking: train encoder --> score each prompt --> select top-k.
Other query dependent methods such as the method presented in https://arxiv.org/abs/2404.02717 could also be compared to.

DPP underperforms on its training domain: On MATH test set with k=4, DPP loses to both BEST-OF-4 and RANDOM-4.
I found this somewhat concerning.
The method performs worse on the very domain it was trained on.
This could hint at overfitting.

Authors only provide limited evidence that the DPP is needed.
There is no ablation study explicitly targeting whether the DPP diversity mechanism contributes beyond a simple supervised ranking mechanism.
A simple baseline of select top-k prompts by learned quality scores maybe could achieve similar performance.

I believe computational cost is not represented correctly.
The authors claim small overhead but (as I understand it) DPP requires also passes through the frozen base model to extract hidden states and running the encoder on the prompt-question pairs and k generations.
This does not seem negligible.
Could the authors provide wallclock times for some experiments?

Experiments in the main paper lack statistical analyses.
There are no error bars, confidence intervals, or significance tests.
It is unclear if results are averaged over multiple runs (and if multiple runs were conducted).
Typical run-to-run performance variation in few-shot settings are 0.25-1.5 accuracy points (for datasets of the size used in the paper).
Most improvements in Table 1 are less than 2 points.
Without multiple runs and statistical analysis, we cannot know if differences are real or noise.
Performance of DPP and competitors is mostly similar (I assume within SEs if multiple runs would be performed and means and SEs would be reported), except for the APPS task where DPP outperforms.
Averaging across tasks (Table 1) is not sensible given different base difficulties of tasks.
Looking into the Appendix F I was quite surprised to see that this appendix now includes quite sophisticated statistical analyses (fixed-effects logistic regression and hypothesis testing).
But the hypothesis test show that the mechanism underlying the method barely works for MATH (small coefficient; marginally significant), in contrast to APPS (stronger coefficient).
For me, this creates some inconsistency:
If the authors are proficient in advanced statistical testing (Appendix F), why are the main results not analyzed with similar efforts?
Formal tests comparing DPP to baselines across datasets with proper mixed models (over multiple runs) are also applicable here.
Also, I would refrain from stating in Appendix F that the authors "propose a statistical test to validate this hypothesis".
The authors rely on standard GLMM machinery.
I admit that not everyone in the ML/DL community will have such a background (and the authors do a good job in referencing the literature) - but after all, from a technical point of view, this is a direct application of long existing and widely used statistical techniques.

**Questions:**

Can you provide the ablation of train encoder to predict quality --> select top-k by score (no DPP).
I am not convinced that DPP selection meaningfully outperform this.

Do you have any hypothesis why DPP-4 underperforms BEST-OF-4 (and RANDOM-4) on MATH?

Can you empirically validate correlated failures? This is the main motivation / justification for the method. But is is not convincingly demonstrated in the paper.

Could you provide actual wall-clock inference times for DPP?

Can you clarify if multiple runs were performed for the main experiments? Can you provide error bars from repeated runs and statistical analysis.

In Figures 2 and 3 why are BEST-OF-K and RANDOM-K shown as horizontal lines?
Are you only plotting single k values for baselines while varying k for DPP?
If so, I find this confusing.
Can you provide curves showing all methods across k=1,2,...?

I would be interested to see a discussion of how does this compare to TEMPERA? TEMPERA achieves strong results with k=1 (in their experiments).

For each test question it would also be interesting to see the oracle upper bound of performance, i.e., what is the maximum pass@k achievable across all possible subsets of prompts. Could this be provided?

Could you discuss distribution shift? How often do we expect the encoder to be retrained in a practical setting?

Do you have any pointers how sensitive the method is to the prompt library quality and its size?

l202 it seems that the appendix reference is broken.

Can you discuss any potential connections to submodular optimization here?
I expect some marginal gain naturally arising when adding a prompt to a set of prompts (depending on the objecgive).
Can't we define a submodular optimization problem that trade-offs quality and diversity of the prompt set directly and simply use greedy techniques?

---

### Official Review · Reviewer_TisX · 2025-11-01

**Soundness:** 2
**Presentation:** 3
**Contribution:** 3
**Rating:** 4
**Confidence:** 2

**Summary:**

This paper proposes a method to improve the reliability of Large Language Models (LLMs) by focusing on input prompt diversification rather than output diversification. Existing approaches like Best-of-k or Random-k generate multiple outputs from the same prompt, often leading to correlated failures. To solve this, the authors train a Determinantal Point Process (DPP)-based prompt selector that simultaneously considers both quality and diversity based on the query. A lightweight encoder generates representations for each query-prompt pair, which are used to construct a DPP kernel $L = F^\top F$ to sample diverse prompt subsets. During training, the model minimizes the KL divergence between the predicted kernel and a target kernel (which combines ground-truth labels and hidden similarities), approximated using unbiased importance sampling. Experiments on MATH, APPS, and WildGuard showed that even with a small budget (k=4), DPP selection achieved higher pass@k and recall@k than Random-k or Best-of-k. It demonstrated particularly efficient reliability improvements for smaller models (1.5B).

**Strengths:**

The strength of this paper lies in its clear problem definition and motivation. The authors noted that small and medium-sized LLMs show little improvement in reliability even with multiple generation attempts (k-sampling) due to "correlated failures." In other words, they identified that existing approaches have a fundamental limitation: they rely solely on output diversity. Instead, they proposed a new perspective: controllably learning input diversity (prompt diversity). To achieve this, they introduced a DPP framework that selects "high-quality and mutually different" prompts for each query, inducing the exploration of diverse reasoning paths even within a limited budget. This idea of structural diversification on the input side—as opposed to simple sampling adjustments or model scaling—is evaluated as the study's greatest strength, as it presents a new, scalable axis for improving reliability.

**Weaknesses:**

1. The paper positions the DPP-4 results as "efficient" despite Table 1 showing it underperforms Best-of-16 and Best-of-24 in absolute performance. This claim of efficiency is not rigorously substantiated. The authors do not quantify the total computational cost of their method. Specifically, the following overheads are not measured or accounted for:
    - Encoder Inference Cost: The computational cost required for the lightweight encoder to generate representations for all candidate query-prompt pairs.
    - DPP Sampling Cost: The computational cost of constructing the kernel and performing the DPP sampling.
    - Prompt Generation Overhead: The initial cost of creating the diverse pool of candidate prompts from which the DPP selector chooses.

2. The paper's central hypothesis is that **prompt diversity** is the key mechanism for reducing correlated failures and improving reliability. However, this claim lacks direct empirical validation, as the contribution of *diversity* is never isolated from the contribution of *quality*. The authors should have compared their DPP-based selector (Quality + Diversity) against a "Quality-Only" baseline.

**Questions:**

Please refer to the weakness.

---

### Official Review · Reviewer_udUL · 2025-11-01

**Soundness:** 2
**Presentation:** 1
**Contribution:** 2
**Rating:** 2
**Confidence:** 3

**Summary:**

The paper proposes a novel method for selecting effective, diverse prompts for sampling, with

**Strengths:**

The proposed method is intriguing in theory.

**Weaknesses:**

W1. Table 1 is confusing since the DPP methods only sample 4 prompts while best-of- and random- both sample 4/8/16/24 prompts. This does not show the result in a positive light, but more importantly, I wonder why the authors chose not to show DPP with more prompts selected, since the potential performance gain is significant.

W2. The ablation study in Section 5.2 has no real experiments. There's also no examination of the empirical approximation of the target $L_i$, or the proposed increased utility or diversity of prompts, etc.

W3. There are many problems in writing for Section 3 and Section 4. For Section 3:
    1. The Notations segment at the very start has several issues. Since Section 3.1 is only dealing with DPPs, the LLM-adjacent notations are not required and will be better to appear later in Section 4. Many things are out of context. DPP is also mentioned before its definition. I would suggest not having the paragraph at all, and just define things sporadically as they appear.
    2. Since DPP is a known concept that may not be readily farmiliar to the machine learning community, it is important to highlight its properties that are crucial to understanding its utility. I'm not an expert on this by any means and I'm basing this on what mathematical understandings I gained from reading this paper, but I think things like the probabilities of $P(\tilde{Y} = Y)$ adding up to one, the correspondence between matrices $L$ and $K$, and what the value in each entry indicate (e.g. higher diagnal value -> higher probabiliy of selection), are important to understanding the mechanism behind the design.
    3. Section 3.2 seems unnecessary. Theorem 4 doesn't suggest that learning DPPs are hard, but only that the sign of nondiagonal elements can be flipped with N degrees of freedom. This can be easily mitigated by fixing the signs somehow, which seems to be the approach taken for the construction of $\hat{L}$ on Lines 258 & 259. There's also an unspecified appendix ?? on Line 202.

W4. For Section 4:
    1. Section 4.1 does not introduce the target kernel, thus the goal of the encoder $\phi$ is quite unclear first time reading. The Gram matrix $F_i^\top F_i$ is supposed to be a low rank approximation (since $d<<M$ I assume) of target $L_i$, but this is not explicitly stated.
    2. The construction of $L_i$ is not fully justified. I understand the intuition of balancing diversity through $K$ and usefulness through the accuracy labels $a_{im}$, but the whole process of Gaussian kernel constructed $\hat{L}$ -> marginal kernel $\hat{K}$ -> weighted average with 0/1 diagnoal -> back to L-ensemble seems to have many arbitrary choices.
    3. I'm not convinced that the encoder can approximate the target DPP well, despite the loss promoting the approximation. See Q4 for more details.

**Questions:**

Q1. I don't think the claim on Line 179/180 that $P(Y \subset \tilde{Y}) = det(K_Y)$ holds. Can you double-check?

Q2. On Line 246/247, does sampling "a subset S of size k from DDP(L)" require taking a conditional distribution on size=k?

Q3. Pertaining to W4.2, can you justify the design choices of $L_i$ further? What, if any, are the theoretical guarantees of this construction?

Q4. Continuing W4.3, One issue is the variance present in the sampling of $Y_{i,b}$'s, as only a small portion of the $2^M$ subsets are sampled at each round, causing a potentially high variance in the loss. Another issue is the aforementioned low rank property of the Gram matrix. Does the inherent rank difference hinder the ability to approximate a DPP?

Q5. Pertaining to W1, why did you not report DPP results with more prompts selected?

---

### Author Response · Authors · 2025-12-03

Dear Reviewers,

We sincerely appreciate your constructive and detailed comments. Your feedback will greatly contribute to improving the rigor, clarity, and overall quality of this work. We will carefully consider your suggestions and further improve our work.

Best Regards,

Authors

---

### Meta-Review · Area_Chair_gNcV · 2026-01-02

**Summary:**

This paper proposes a method for diversifying input prompts using determinantal point process (DPP) sampling guided by an accuracy–diversity target kernel, where diversity is measured via similarities between hidden activations. A lightweight encoder is trained to approximate this target kernel by minimizing a KL-divergence objective using an unbiased gradient estimator, and is used at inference time to sample a diverse subset of prompts for each query.

**Reviewer Concerns:**

Reviewer udUL
* Confusing Table 1 (DPP is evaluated only with 4 prompts, while baselines are tested with 4/8/16/24 prompts).
* The ablation study in Section 5.2 contains no real experiments.
* Presentation issues.


Reviewer TisX
* The claim that DPP-4 is “efficient” is not supported by the results.
* Questions the central assumption that prompt diversity leads to improved accuracy and reliability.


Reviewer A6pk
* The central claim that baselines suffer from correlated failures (especially with small LLMs) is not empirically demonstrated or supported by existing work.
* Baselines are training-free, leading to an unfair comparison.
* DPP underperforms even on its training domain.
* Limited empirical support for the claimed advantages of the proposed method.


Reviewer GiWc
* Skepticism about generalization beyond the training domain.
* The encoder cannot be considered “lightweight.”
* Scalability issues with respect to (M).
* Concerns about training stability.
* Missing ablations.

**Reviewer Scores:**

I largely agree with the reviewers’ criticisms, and the authors did not participate in the rebuttal. I therefore recommend rejection.

---

### Decision · Program_Chairs · 2026-01-26

Reject